# Identification of Novel Biomarker for Early Detection of Diabetic Nephropathy

**DOI:** 10.3390/biomedicines9050457

**Published:** 2021-04-22

**Authors:** Kyeong-Seok Kim, Jin-Sol Lee, Jae-Hyeon Park, Eun-Young Lee, Jong-Seok Moon, Sang-Kyu Lee, Jong-Sil Lee, Jung-Hwan Kim, Hyung-Sik Kim

**Affiliations:** 1School of Pharmacy, Sungkyunkwan University, Suwon 16419, Korea; caion123@nate.com (K.-S.K.); jinsol3361@skku.edu (J.-S.L.); sky3640@skku.edu (J.-H.P.); 2Department of Pharmacology, Institute of Health Sciences, College of Medicine, Gyeongsang National University, Jinju 52727, Korea; 3Department of Internal Medicine, Soonchunhyang University Cheonan Hospital, Cheonan 31151, Korea; eylee@sch.ac.kr; 4BK21 Four Project, College of Medicine, Soonchunhyang University, Cheonan 31151, Korea; 5Institute of Tissue Regeneration, College of Medicine, Soonchunhyang University, Cheonan 31151, Korea; 6Department of Integrated Biomedical Science, Soonchunhyang Institute of Medi-Bio Science, Soonchunhyang University, Cheonan 31151, Korea; jongseok81@sch.ac.kr; 7BK21 Plus KNU Multi-Omics Based Creative Drug Research Team, College of Pharmacy, Kyungpook National University, Daegu 41566, Korea; sangkyu@knu.ac.kr; 8Department of Pathology, Institute of Health Sciences, College of Medicine, Gyeongsang National University Hospital, Jinju 52727, Korea; jongsil25@gnu.ac.kr; 9Department of Convergence Medical Science, Gyeongsang National University, Jinju 52727, Korea

**Keywords:** RNA-sequencing, biomarker, urine, CFD, CXCR6, C4b, LIF, diabetic nephropathy

## Abstract

Diabetic nephropathy (DN) is one of the most common complications of diabetes mellitus. After development of DN, patients will progress to end-stage renal disease, which is associated with high morbidity and mortality. Here, we developed early-stage diagnostic biomarkers to detect DN as a strategy for DN intervention. For the DN model, Zucker diabetic fatty rats were used for DN phenotyping. The results revealed that DN rats showed significantly increased blood glucose, blood urea nitrogen (BUN), and serum creatinine levels, accompanied by severe kidney injury, fibrosis and microstructural changes. In addition, DN rats showed significantly increased urinary excretion of kidney injury molecule-1 (KIM-1) and neutrophil gelatinase-associated lipocalin (NGAL). Transcriptome analysis revealed that new DN biomarkers, such as complementary component 4b (C4b), complementary factor D (CFD), C-X-C motif chemokine receptor 6 (CXCR6), and leukemia inhibitory factor (LIF) were identified. Furthermore, they were found in the urine of patients with DN. Since these biomarkers were detected in the urine and kidney of DN rats and urine of diabetic patients, the selected markers could be used as early diagnosis biomarkers for chronic diabetic nephropathy.

## 1. Introduction

Diabetes mellitus (DM) is one of the most common and complex metabolic disorders. The disease represents one of the greatest medical and socioeconomic challenges throughout the world due to the high rates of diabetic complications and mortality associated with the disease [1]. DM causes high blood glucose levels and is one of the global health burdens owing to its increasing prevalence, and is expected to affect 495 million people [1,2].

Diabetic nephropathy (DN) is one of the most devastating microvascular complications of DM, developing in nearly one-third of patients with type 1 or type 2 diabetes [3]. It is by far the most common cause of chronic kidney disease worldwide, frequently leading to end-stage renal disease (ESRD) and the need for renal replacement therapy [4]. Diabetes and diabetic complications, such as diabetic cardiomyopathy, neuropathy, and kidney disease are accompanied by mitochondrial oxidative damage and dysfunction [3].

Diagnostic markers to detect DN at early stage are important as early intervention can slow the loss of kidney function and reduce adverse outcomes. The appearance of small amounts of protein albumin in urine, called microalbuminuria has been accepted as the earliest marker for development of DN. However, it has been reported that a large proportion of renal impairment occurs even before appearance of microalbuminuria [5]. Albuminuria has several confounding issues associated with it such as exercise, urinary tract infection, acute illness, and cardiac failure. Furthermore, it has been reported to occur in the urine of normal-diet subjects, indicating the non-specificity of albuminuria for accurate prediction of diabetic kidney disorder [6].

High glucose levels in blood vessels play a crucial role in the progression of DN. Hyperglycemia induces metabolic dysfunctions in mitochondria and the glucose metabolic pathway through overproduction of reactive oxygen species (ROS) [7]. High concentration of glucose leads to formation of covalent adducts with plasma proteins through glycation. One of these events is formation of advanced glycation end products (AGEs) and these are a major risk factor for diabetic complications [8]. Podocytes, an important component of the glomerular filtration barrier, can become abnormal after chronic exposure to hyperglycemia. The loss of podocytes is one of the earliest glomerular morphologic changes and has an important role in development of DN. DN is clinically characterized by proteinuria and impaired kidney function in diabetic patients [9,10,11]. Pathologically, DN patients usually have kidney hypertrophy, basement membrane thickening, deposition of extracellular matrix proteins, glomerular sclerosis, and interstitial fibrosis [12]. Although DN patients can be managed by control of blood glucose and blood pressure, many patients eventually progress to renal failure [13,14]. Hence, understanding the pathophysiology of DN and development of new biomarkers will be of high demand for early diagnosis of DN.

Biomarkers are considered as biological entities which are used to indicate the states of healthy or diseased cells, tissues, or individuals. From the clinical perspective, biomarkers have a substantial impact on the care of patients who are suspected to have disease, or those who have or have no apparent disease [15]. Currently, biomarkers are applied at a molecular level such as genes, proteins, metabolites, glycans, and other molecules, which can be used for diagnosis of diseases, prognosis, prediction of therapeutic responses, as well as therapeutic development for many diseases [16,17,18,19,20,21,22,23].

In this study, we evaluated differences in gene expression between the normal-dieted group and high fat-dieted DN rat model at various ages (5- and 12-week-high fat induced Zucker diabetic fatty rat) using RNA-Seq and discovered new biomarkers for diabetic nephropathy.

## 2. Materials and Methods

### 2.1. Chemicals and Reagents

The primary antibodies for anti-LIF, anti-KIM-1, and anti-NGAL were purchased from Abcam (Cambridge, MA, USA). Anti-C4b antibody was purchased from OriGene (Rockville, MD, USA). Anti-CFD and anti-CXCR6 antibody was purchased from MyBioSource (San Diego, CA, USA). Horseradish peroxidases (HRP)-conjugated secondary antibody were purchased from Santa Cruz Biotechnology (Santa Cruz, CA, USA). Hematoxylin, eosin, and periodic acid-Schiff (PAS) stain kit were purchased from Dako (Glostrup, Denmark).

### 2.2. Animal Experiments

Four-week-old male Zucker diabetic fatty rats (Lepr(fa/fa)) were purchased from Central Lab Animal Inc. (Seoul, Korea). They were housed under controlled temperature (23 ± 2 °C) and a 12 h light/dark cycle in filtered-air laminar-flow cabinets, and handled using aseptic procedures. All procedures were approved by the institutional animal care committee of Sungkyunkwan University (code, SKKUIACUC2019-03-25-1; April 2019). The rats were randomly divided into three groups: normal-diet group (ND, *n* = 3) and high-fat diet (HFD) induced diabetic groups, 5-week high-fat fed group (5 W, *n* = 4) and 12-week high-fat fed group (12 W, *n* = 4). The normal diet (ND) group received only the normal diet free from cholesterol. The HFD, comprising proteins, fats, carbohydrates, fibers, minerals, and vitamins with 60 kcal% fat, was purchased from Research Diets, Inc. (New Brunswick, NJ, USA), and was fed to the rats in order to indicate a time point for inducing diabetic nephropathy (Figure 1A). The blood glucose level was measured by using a glucometer (ACCU-CHEK Performa). The blood glucose levels and body weights were estimated every week. Urine samples were collected every week. Each animal was kept in the metabolic cage overnight, and 24-h urine samples were collected from individual animals into containers (0.1% sodium azide). Urine samples were immediately centrifuged at 4000 rpm for 15 min and stored in aliquots at −80 °C for subsequent analysis [24]. Before being sacrificed, all animals were fasted for 12 h before completing the experiment.

All the rats were anesthetized with CO_2_ gas, and blood was collected from abdominal aorta. The blood samples were centrifuged at 3000 rpm for 10 min and stored at −80 °C until further use. After sacrificing the rats, their major organs (liver, kidney, pancreas, and testis) were perfused with normal saline for removing the residual blood, and stored at −80 °C for experiments.

### 2.3. Analysis of Serum and Urine Biochemical Parameters

Blood was collected from the abdominal aorta and collected in 15-mL plain tubes and lavender-top (EDTA) specimen tubes for analysis of hemoglobin A1C (HbA1C). Within 1 h of collection, plain tube blood samples were then centrifuged at 3000 rpm for 10 min to collect serum. The sera were immediately stored at −80 °C to analyze blood urea nitrogen (BUN), serum creatine (sCr), glucose levels, aspartate aminotransferase (AST), and alanine aminotransferase (ALT) activities, by using an Olympus AU400 chemistry analyzer (Tokyo, Japan). Advanced glycation end products (AGEs) using the VetScan analyzer (Abaxis Inc., Union City, CA, USA). The levels of urinary albumin, creatinine, and protein were estimated by using a Hitachi 7180 auto analyzer (Hitachi, Tokyo, Japan).

### 2.4. Oral Glucose Tolerance Test (GTT)

Prior to the test, the rats were fasted for 16 h and transferred to private cages. Blood was obtained from a tail cut (by removing the distal 2 mm of the tail) and was assessed for baseline glucose levels by using a glucometer (ACCU-CHEK Performa). The rat then received 2 g/kg body weight of a 100 mg/mL glucose solution (Sigma, St. Louis, MO, USA) #G8769) in sterile water delivered by oral gavage. At 15, 30, 60, and 120 min after the administration of glucose, blood was collected to measure the glucose concentration.

### 2.5. Histology

The kidney tissue was preserved immediately after collection in 10% neutral buffered formalin (NBF). The paraffin sections of these tissues (4 μm) were prepared for hematoxylin and eosin (H&E) staining for determining the morphological abnormalities of the tissues. Additionally, periodic acid-Schiff (PAS) stain was performed for detection of accumulated glycogen in the kidney. At least 5 random positive areas in each section were photo-imaged at 200× magnification.

### 2.6. Transcriptomics

The total RNA was extracted from each kidney tissue and after qualification and quantification, their mRNAs were enriched by poly-T oligo-attached magnetic beads and fragmented by using divalent cations in NEBNext First Strand Synthesis Reaction Buffer (5×), followed by purification. Subsequently, the cDNA was synthesized using random hexamer primer and M-MuLV Reverse Transcriptase (RNase H-), DNA Polymerase I, and RNase H. Sequencing libraries were prepared by using TruSeq Stranded mRNA LT Sample Prep Kit for Illumina (NEB, Ipswich, MA, USA) according to manufacturer’s instructions and the library was assessed using Agilent 2100 bioanalyzer. Finally, the library was sequenced on an Illumina Novaseq 6000 platform and yielded 150 bp paired-end reads. The generated raw reads were evaluated by the CASAVA base and after removing adapter or ploy-N and low-quality reads, the remaining high quality of reads were evaluated for their Q30, GC content, and alignment efficiency.

In order to more accurately evaluate the gene expression changes as diabetes progresses in the kidney, the differentially expressed genes (DEGs) were selected based on the following criteria: compared to the normal diet group, DEGs with up-, down-regulated expression in 5-week high-fat diet (5 W), and 12-week high-fat diet (12 W) fed groups. The selected DEGs were analyzed by gene clustering, Gene Ontology (GO) (http://www.geneontology.org, accessed on 1 June 2019).) and Kyoto Encyclopedia of Genes and Genomes (KEGG) enrichment (http://www.genome.jp/kegg/, accessed on 1 June 2019). GO analysis was performed to construct the main function of the differentially expressed mRNAs (log 2 |f.c| > 1.5, *p*-value < 0.05). The −log 10 (*p*-value) values with the enrichment score represents the significance of the GO term. KEGG pathway analysis was performed to harvest pathway clusters covering differentially regulated mRNA profiles in the molecular interaction networks. The −log 10 (*p*-value) values with the enrichment score indicates the significance of correlation in the pathway. The genes in the enriched biological pathways (*p* < 0.05) were chosen and Cytoscape was used to construct the pathway network.

### 2.7. Western Blot Analysis

Urine and conditioned media were prepared approximately 1 h before analysis; thereafter, it was vortexed and centrifuged at 3000× *g* to filter out impurities. Protein concentrations were measured using a protein assay kit (Bio-Rad, Hercules, CA, USA) according to the manufacturer’s instructions. Equal amounts of protein were loaded onto 6–15% SDS-PAGE gel. After electrophoresis, the gels were transferred to a polyvinylidene difluoride (PVDF) membrane (Millipore, Billerica, MA, USA). After transfer, the membranes were blocked at 25 °C for 1 h using TNT buffer (10 mM Tris-Cl, pH 7.6, 100 mM NaCl, and 0.5% Tween 20) containing 5% skim milk. Then, the membrane was incubated overnight with primary antibodies at 4 °C and washed twice, for 5 min each time. Subsequently, membranes were incubated with anti-goat IgG (1:10,000) or anti-rabbit IgG (1:10,000) horseradish peroxidase (HRP)-conjugate for 60 min at room temperature. Membranes were washed twice for 10 min with PBS-Tween 20. Finally, the blots were developed by using an enhanced chemiluminescence (ECL)-plus kit (Amersham Biosciences, Amersham, Buckinghamshire, UK).

### 2.8. RNA Extraction and Quantitative Real-Time PCR

Total RNA was extracted from kidney tissue using Qiazol Lysis Reagent (Qiagen, Germantown, MD, USA) according to the manufacturer’s protocol. After RNA quantification, cDNA was synthesized by reverse transcription using the Reverse Transcription Master Premix (Elpis Biotech, Daejeon, Korea). Quantitative RT-PCR was performed on a LightCycler 96 Real-Time PCR system (Roche, Basel, Switzerland) using the FastStart Essential DNA Green Master (Roche). 2-ΔΔCt values were calculated to obtain relative fold expression levels. The sequence of primers used in the study are listed in Appendix A.

### 2.9. Clinical Urine Sample Analysis from DN Patients

Urine samples from six normal and thirteen patients were collected at Soonchunhyang University Hospital (Cheonan, Korea) from June 2016 to July 2016 to detect candidate biomarkers (C4B, CFD, CXCR6, LIF, NGAL, and KIM-1). Urine samples of 10 mL were collected from all subjects, and then, urine was immediately centrifuged at 1000× *g* for 5 min; supernatants were stored at −80 °C prior to analysis. Urines were subjected to Western blot analysis. The patients provided written informed consent, and collection of human urine samples was approved by the Medical Ethics Committee of Soonchunhyang University, School of Medicine and Hospital, Cheonan Korea (SCHCA 2015-12-023).

### 2.10. Statistical Methods

The experimental data were analyzed at least three times, and are expressed as the mean ± standard error (SE) or median, minimum, and maximum. For the determination of statistically significance, we used a one-way analysis of variance (ANOVA) and then compared it with Bonferroni’s multiple comparison tests. * *p* < 0.05 and ** *p* < 0.01 indicate the significant differences between the control and the treatment groups. All of the statistical comparisons were performed using Sigma Plot graphing software and the Statistical Package for the Social Sciences v.13 (SPSS, Inc., Chicago, IL, USA).

## 3. Results

### 3.1. Effects of Diabetes Progression on Body Weight and Organ Weight Change in Rats

Over the experimental period, all animals were weighed every week. The high-fat diet-induced diabetes group showed significantly increased body weight compared to the normal diet group. The kidney and liver weights were significantly increased compared to the normal-diet group. Testis and pancreas weights were reduced compared with normal-diet group (Figure 1B). These results were similar to a previously reported data that hyperglycemic conditions cause testicular morphological changes and induce apoptosis [25] and diabetes causes morphological changes in pancreas and weight loss [26].

### 3.2. Serum and Urinary Biochemical Parameters Changes as Diabetes Progression

Biochemical analysis was performed to assess organ toxicity induced by diabetes progression using urine and serum. Serum biomarkers of liver injury (AST, ALT) and renal injury (BUN, sCr) levels were significantly increased at the indicated time point compared to the normal diet group. Measurement of glycated hemoglobin (HbA1c), advanced glycation end products (AGEs), and glucose concentration were performed to diagnose diabetes mellitus. Serum HbA1c, AGEs, and glucose levels were increased in time-dependent analysis. As a result of oral glucose tolerance test (oGTT), blood sugar recovery level was decreased as the induction period of diabetes increased (Figure 2A). Urinary creatinine, albumin, and protein concentration were significantly increased as diabetes progressed (Figure 2B). These results indicated the presence of severe kidney injury in diabetic rats. Furthermore, histopathological evaluation was performed to more clearly confirm diabetes induced renal injury.

### 3.3. Histological Examination

For histopathological examination, the proximal tubules and glomerular damages were visualized using H&E staining (Figure 3). In the normal diet group, normal morphology in the glomeruli and the proximal tubules, as well as the vessels, were clearly shown in the kidneys of rats. However, glomerular and tubular injuries such as glomerular hyaline droplet, increased mesangial matrix, basophilic renal tubule, hyaline cast, and tubular dilatation were observed in the diabetes induced rat groups, its severity increased time dependently. Mesangial expansion, hyaline cast, and hyaline droplet are well known as diabetic nephropathy pathological findings. Periodic acid-Schiff (PAS) staining was performed to detect the polysaccharides such as glycogen and glycoproteins in kidneys. PAS positive glycogens (black arrow) were significantly increased in diabetes-induced rat kidneys. Based on these results, kidney injury was evidently identified in the high-fat diet-induced ZDF rat DN model.

### 3.4. Identification of Differentially Expressed Genes during Progression of DN

Transcriptome analysis was performed to confirm the changes in gene expression according to the progression of diabetes induced kidney injury at the indicated time points (5 W, and 12 W). To identify candidate biomarkers in progressive diabetes induced nephropathy, three rats were randomly selected from the 5-week group (5 W) and 12-week group (12 W) in diabetes induced rats, and another three from the 12-week group were randomly selected from the normal diet group (ND). The kidney tissues were preserved in liquid nitrogen and total RNAs were used for whole transcriptome analysis with RNA-Seq. Generated raw reads were evaluated by the CASAVA base and after removing adapter or ploy-N and low-quality reads, the remaining about 10.64 GB highly quality reads were evaluated for their Q30, GC content, and alignment efficiency. The sequenced reads had the alignment efficacies of 96.2–96.5%. Before starting the DEG analysis, multidimensional scaling analysis (MDS) was performed to estimate the variance of individual samples in each group (data not shown).

The analysis of bioinformatics was divided into two independent comparisons: 5 weeks vs. normal diet control (5 W vs. ND) and 12 weeks vs. normal diet control (12 W vs. ND). Volcano plots provided an overview of the differential expression of mRNAs. An absolute log 2 FC cutoff value of 2 was utilized as the criteria to confirm up-regulated and down-regulated mRNAs (FDR < 0.05) (Figure 4A,B).

The mRNA-sequencing reveled 2427 differentially expressed genes (DEGs; log 2 |fc| ≥ 1.5) between the diabetes induced group (5 W, 12 W) and normal-diet (ND) group. An absolute log 2 FC cutoff value of 1.5 was utilized as the criteria to confirm up-regulated and down-regulated mRNAs. Using this standard, there were 154 up-regulated mRNAs and 41 down-regulated mRNAs in 5 W vs. ND, and 630 up-regulated mRNAs and 216 down-regulated mRNAs in 12 W vs. ND were verified. Venn diagrams revealed possible relations between 5 W vs. ND and 12 W vs. ND. In total, 58 mRNAs and 709 mRNA were significantly expressed in 5 W and 12 W, respectively when compared to each control (ND). (Figure 4C,D). Heatmaps showed the common changes of mRNA expression between 5 W vs. ND and 12 W vs. ND (Figure 4E,F) (FDR < 0.05).

### 3.5. The Altered mRNAs Were Enriched in Certain Biological Functions

GO and KEGG pathway enrichment were displayed to verify the biological implications and functional analysis [27]. We associated differentially expressed genes (DEGs) in 5 W vs. ND and 12 W vs. ND with three modules provided by GO. The GO analysis performed further confirmed that the mRNAs with up-regulated expression were enriched in biological processes, included cell cycle (5 W) and immune system process (12 W). However, the down-regulated expressions were enriched in biological processes, including complement activation (lectin pathway) (5 W) and the organic acid metabolism process (12 W).

Next, using KEGG analysis, the differentially regulated pathways were enriched as up-regulated or down-regulated. Our data showed that the top 5 pathways correlate to up- and down-regulated mRNAs for 5 W vs. ND and 12 W vs. ND. Enrichment analysis revealed that the cytokine-cytokine receptor interaction and cell cycle were the primary pathways among the significantly up-regulated mRNAs for 5 W vs. ND and 12 W vs. ND (Appendix A). The top enriched pathway for down-regulated transcripts in 5 W vs. ND was steroid hormone synthesis (Appendix A), whereas metabolic pathway was the top pathway in 12 W vs. ND (Appendix A).

### 3.6. Identification of Secreted Genes Closely Related to Progressive Kidney Injury in Diabetes Using DEGs

These GO and KEGG pathway results have been shown to be associated with toxicological phenotypes of inflammation and fibrosis induced by diabetes progression in diabetic kidneys. Among the up-regulated genes involved in these Gene Ontology Biological Processes (GOBPs), we next focused on the following secretory and regulatory factors that can modulate renal failure. To identify key regulatory factors among these mRNAs, we collected mRNA expression profiles from NCBI GEO databases generated from renal injury including acute kidney injury, chronic kidney injury, diabetic nephropathy, nephritis, and end stage renal disease. From these profiles, we identified 91 DEGs showing increased mRNA expression in DN kidney compared to the normal diet groups (Appendix A). In the previous study, development of diabetic nephropathy related pathways is grouped into four main categories: either metabolic factors, hemodynamic factors, growth factors, cytokines and intracellular factors, or a combination [28]. In addition to these four pathways, the cytokine-mediated signaling pathway and the innate immune system, including the complement system play an important role in development of diabetic nephropathy [29,30,31].

Based on these results, we selected the candidate biomarkers that are associated with kidney injury and also related to the complement system and cytokine mediate signaling pathway.

### 3.7. Validation of Candidate Biomarkers

In the current study, complementary systems and cytokine mediate signaling pathway were major regulatory factors in diabetes-induced nephrotoxicity using transcriptomic and proteomic studies. Thus, we selected the complementary complement 4b (C4b), complement factor D (CFD), C-X-C motif Chemokine Receptor 6 (CXCR6), and leukemia inhibitory factor (LIF) as candidate biomarkers for early detection of diabetic nephropathy. To validate whether progression of diabetic nephropathy increased expression levels of candidate biomarkers, immunoblot analyses were performed with in vivo and clinical samples. We next evaluated the potential of C4b, CFD, CXCR6, and LIF as biomarker candidates for the assessment of nephrotoxicity in vivo. qRT-PCR were performed to evaluate the RNA-Seq results. Kidney mRNA expression levels increased time-dependently as diabetes progressed (Figure 5A). The urinary excretion of CFD, CXCR6, LIF, and C4B were increased in rats in a time-dependent manner (Figure 5B). A previous study indicated that excretion levels of protein-based biomarkers including urinary kidney injury molecule-1 (KIM-1), and neutrophil gelatinase-associated lipocalin (NGAL) exhibit high sensitivity for the detection of renal injury [32]. Similarly, our study also observed that the urinary excretion of these biomarkers increased in a dose-dependent manner.

The reliability of C4b, CFD, CXCR6, and LIF as a biomarker for detecting diabetic nephropathy was investigated in clinical trials. The urinary levels of C4b, CFD, CXCR6, LIF, KIM-1, and NGAL were markedly increased in diabetic nephropathy patients compared to normal subjects (Figure 6). These results suggested that C4b, CFD, CXCR6, and LIF may be considered as a noninvasive biomarker for early detection of diabetic nephropathy in a clinical trial.

## 4. Discussion

Diabetic nephropathy is one of the microvascular complications of diabetes and is the leading cause of end-stage renal disease (ESRD). Diabetic nephropathy affects approximately 20–40% of patients with the prevalence of type 2 diabetes. Despite current therapies, there is large residual risk of diabetic kidney diseases onset and progression [33,34,35]. Therefore, development of new biomarkers for early detection of diabetic nephropathy is needed to improve health outcomes including reducing morbidity and mortality from diabetes complications.

Whole-transcriptome sequencing and bioinformatics analysis approaches have contributed to identification of sensitive and reliable biomarkers for early detection and assessment of diabetes induced nephrotoxicity. These findings revealed the pathways involved in dysregulated genes and the biological implications involved in the progression of diabetic nephropathy in diabetic rat models. Moreover, we performed a time series clustering analysis to identify significant differentially expressed genes. Our data revealed that a total of 2427 mRNAs were differentially expressed in the kidney, accompanied by variation in expression over time associated with DN progression.

Pathway enrichment analysis results have indicated that identified genes involve a wide range of classical biological processes, including cell cycle, immune response, complement activation, metabolic pathway, and others. These pathways contribute to the processes of renal hypertrophy, fibrosis, and inflammation. We identified C4b, CFD, CXCR6, and LIF that were time dependently increased as diabetes progression as non-invasive biomarker candidates for diabetic nephropathy. These candidate biomarkers are related to the complement system and interaction between cytokine and receptors pathways.

The complement system is an important component of the immune system and has a key role in facilitating the clearance of microorganisms and damaged cells by antibodies and phagocytic cells [36]. The evidence for a connection between the complement system and renal dysfunction spans decades, and is supported by findings from experimental and clinical studies [37,38]. Complement systems are associated with many kinds of renal disease such as glomerulonephritis, lupus nephritis, IgA-nephropathy, ischemia/reperfusion damage, kidney transplantation, atypical hemolytic-uremic syndrome, and progression of chronic renal diseases [31].

Complement component 4b (C4b), active serine proteases, is a highly reactive molecule that binds to the pathogen surface. In the classical complement pathway, C4b is covalently bonded to a surface molecule of a pathogen, carrying out its role as a platform for the next step in the complementary cascade [39]. Our results showed that expression levels of C4b in rat urine, kidney tissue, and clinical samples were increased by progression of diabetic nephropathy. In a previous study, hyperglycemia led to over activation of complement pathways induced by glycation of complement regulatory proteins, which explain the involvement of complement in the development of diabetic nephropathy [40,41]. The results implicate that the expression level of C4b in urine may be overproduction of uncontrollable complement systems. The role of C4b in progression of diabetic nephropathy is not clearly elucidated. However, urinary C4b level was increased in diabetic nephropathy.

Complement factor D (CFD, or adipsin) is a serine protease that stimulates glucose transport for triglyceride accumulation in fats cells and inhibits lipolysis. CFD is involved in the alternative complement pathway of the complement system where it cleaves factor B [39]. Our results show that expression levels of CFD in rat urine, kidney tissue, and clinical samples were increased by progression of diabetic nephropathy. It is known that plasma concentrations of CFD increase up to 10-fold in the end stage of renal failure, suggesting that its metabolism is related to renal functions [42,43,44]. CFD metabolism in patients showed that the proportion of CFD elimination in the urine was increased in patients with tubular dysfunctions, it demonstrated that under normal circumstances CFD is filtered through the glomeruli and reabsorbed by tubular cells [45]. CFD does not seem to be involved in pathogenesis of diabetic nephropathy. However, it happens to be a protein that has an output trend similar to the 24-h urine protein [46]. Based on this observation and pathological changes of diabetic kidneys, the concentration of CFD changes in urine with diabetic nephropathy can be explained. Urinary CFD is used as an indicator of non-invasive biomarker for diagnosing pre-eclampsia-induced acute kidney injury [47]. However, this study is the first to demonstrate that expression levels of CFD in rat urine, as well as patients’ urine, can be used in diagnosis of diabetic nephropathy.

The activations of inflammatory response in diabetic kidneys are well connected with several factors such as hyperglycemia, renin-angiotensin system, and oxidative stress which results in the infiltration of the organ by monocyte and lymphocytes, and release adverse molecules such as reactive oxygen species and pro-inflammatory cytokines [48,49,50,51]. The leukocytes migration from blood vessels into tissue holds a significant role in the development of renal disorders through triggering cell injury [52,53,54,55].

CXCL16 is a newly invented small CXC chemokine, expressed as transmembrane protein and a cytoplasmic tail with a potential tyrosine phosphorylation [56]. The secreted CXCL16 acts as a chemoattractant that promotes migration of CXCR6 expressing cells and induces IFN-γ and TNF-α [57,58]. C-X-C motif Chemokine Receptor 6 (CXCR6) (STRL33/BONZO/TYMSTR) is the receptor of CXCL16 and expressed on inflamed tissues, such as rheumatoid joints and inflamed livers, and on natural killer T cells, aortic smooth muscle cells, astrocytes, epithelial cells, and stromal cells [59,60,61,62,63,64,65].

Our results revealed that the expression levels of CXCR6 in rat urine, kidney tissue, and clinical samples were up-regulated by development of diabetic nephropathy. Essentially, our finding provides a first-hand clue that CXCR6 was up-regulated in urine during progression of diabetic nephropathy. A previous study demonstrated that the CXCL16 signaling pathway contributes to the renal injury in diabetic nephropathy mouse model [66] and CXCR6 regulates the collagen deposition and expression of collagen I and fibronectin through recruitment of bone marrow derived fibroblast in renal fibrosis [67]. Thus our findings suggest that CXCL16/CXCR6 pathway may contribute to progression of end-stage renal diseases. These findings can be applied to develop a novel clinical strategy for renal fibrosis. However, the role of CXCR6 in progression of diabetic nephropathy is not clearly elucidated. Additional studies are required to explore the underlying mechanism in DN.

Leukemia inhibitory factor (LIF) is a pleiotropic glycoprotein belonging to the interleukin-6 family of cytokines. LIF regulates the nephrogenesis and plays an important role in the protective effect induced by oxidative stress [68,69]. In the present study, we observed that prolonged hyperglycemia caused an increase of LIF in rat urine, kidney tissue, and clinical samples time-dependently. A recent study shows that LIF promotes tubular regeneration after acute renal failure [70]. Although LIF has been studied actively in various organs, including kidney, and shows its anti-apoptotic and regulating functions on various cells, there is little information about the effects of LIF on podocyte apoptosis in diabetic nephropathy [69].

In conclusion, our study provides transcriptomic data of an animal model with progressive diabetes-induced nephrotoxicity, and also demonstrates that C4b, CXCR6, CFD, and LIF are specific and sensitive biomarkers for early detection of diabetic nephropathy. Using diabetic nephropathy models, data demonstrate that urinary levels of C4b, CXCR6, CFD, and LIF were markedly increased compared with conventional biomarkers (KIM-1, NGAL). Further, urinary C4b, CXCR6, CFD, and LIF levels might also potentially be employed as biomarkers to monitor the effectiveness of pharmacological interventions in diabetic nephropathy patients. Based on a preliminary scale of clinical urine tests, it was also found that C4b, CXCR6, CFD, and LIF were demonstrated to be biomarker candidates for early detection of diabetic nephropathy.

## Figures and Tables

**Figure 1 biomedicines-09-00457-f001:**
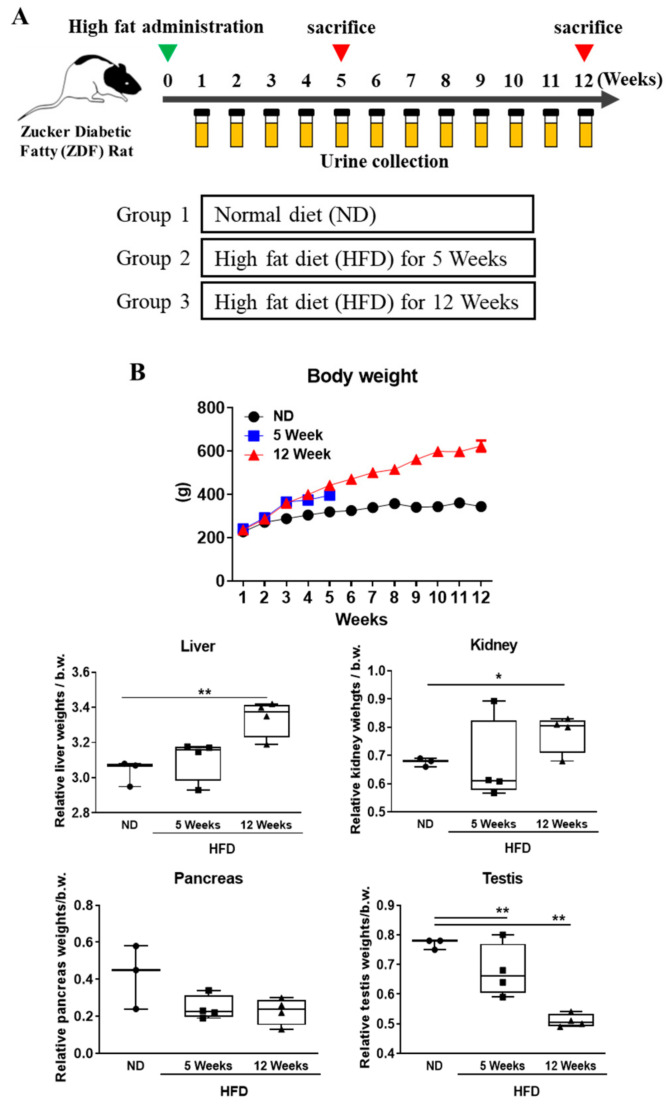
Experimental design and body weight and organ weight changes. (**A**) Zucker diabetic fatty rats were divided into three groups (normal diet group, high fat diet groups (5 and 12 weeks)). Urine samples were collected from all subjects weekly. (**B**) Body, liver, pancreas, and testis weight were measured in each group. All results showed changes in relative organ weight ratio (%) at indicate time point. Data are expressed as the mean ± SD of each group. * *p* < 0.05; ** *p* < 0.01; ND, normal diet; HFD, high-fat diet.

**Figure 2 biomedicines-09-00457-f002:**
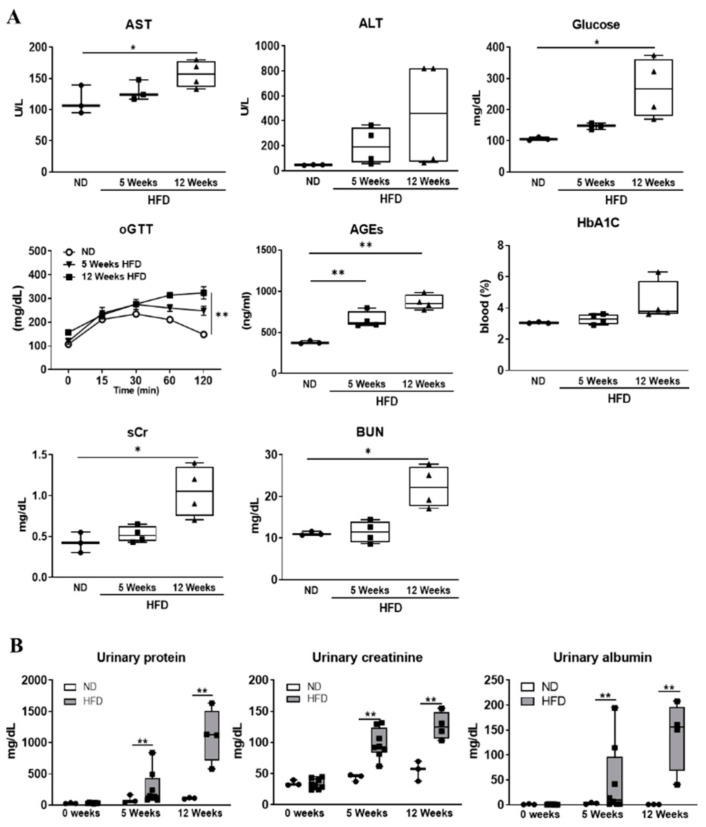
Effect of hyperglycemia on biochemical parameter changes. (**A**) Levels of blood urea nitrogen (BUN), serum creatinine (sCr), glucose, aspartate aminotransferase (AST), alanine aminotransferase (ALT), hemoglobin A1C (HbA1C), glucose, and advanced glycation end products (AGEs) in the blood serum were measured after fasting at the end of experiment. All the above biochemical parameters were analyzed at 12-h fasting state condition. Oral glucose tolerance test (oGTT) was performed to diagnose diabetes. (**B**) Urinary creatinine, albumin, and protein levels estimated at indicated time point. The values represent means ± SEM. * *p* < 0.05; ** *p* < 0.01; ND, normal diet; HFD, high-fat diet.

**Figure 3 biomedicines-09-00457-f003:**
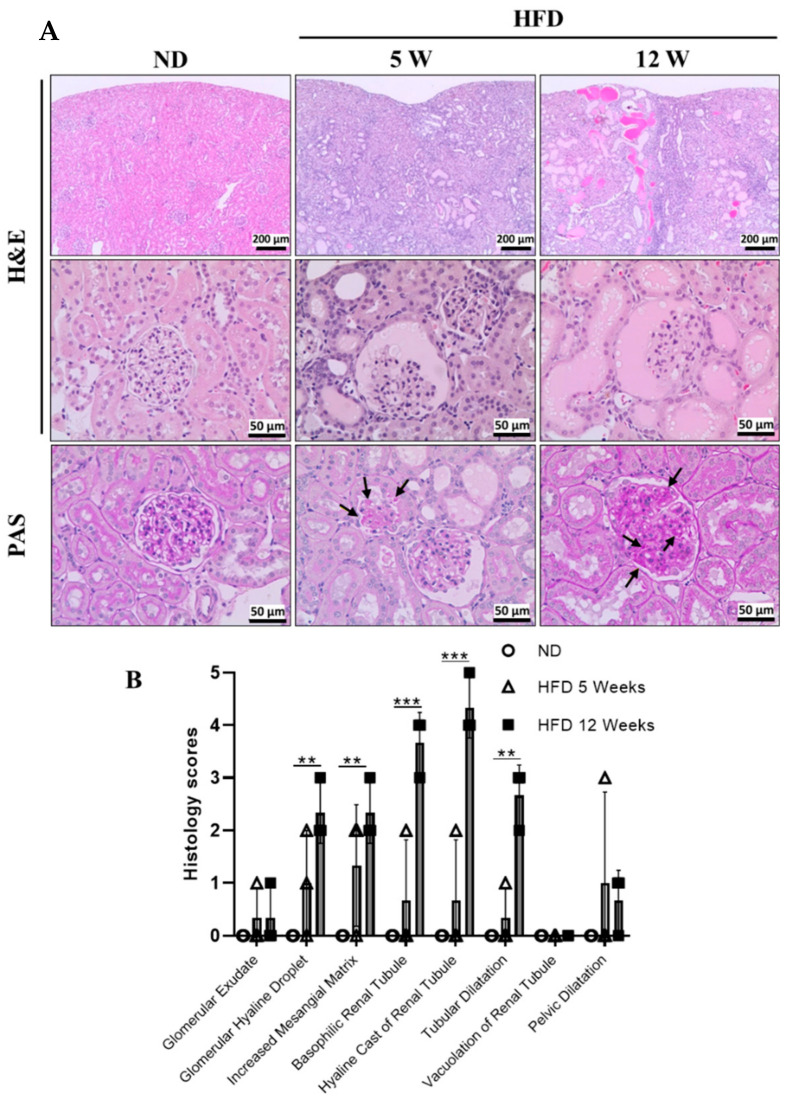
Kidney histology as diabetes progression. (**A**) Kidney samples were subjected to histology examination. Hematoxylin and eosin (H&E) and periodic acid-Schiff (PAS) staining analysis of renal histological abnormalities showing hyaline cast of renal tubules, mesangial cell expansion, thickening of the basement membrane, increased mesangial matrix, formation of glomerular hyaline droplet, and accumulation of PAS positive glycogen (black arrow) stores in kidneys. (**B**) Histology scores for the renal injury were shown. Scores (0, normal; 1, minimal; 2, slight; 3, moderate; 4, marked; 5, severe), ND, normal diet; HFD, high-fat diet; 5 W, 5 weeks; 12 W, 12 weeks. ** *p* < 0.01, *** *p* < 0.001.

**Figure 4 biomedicines-09-00457-f004:**
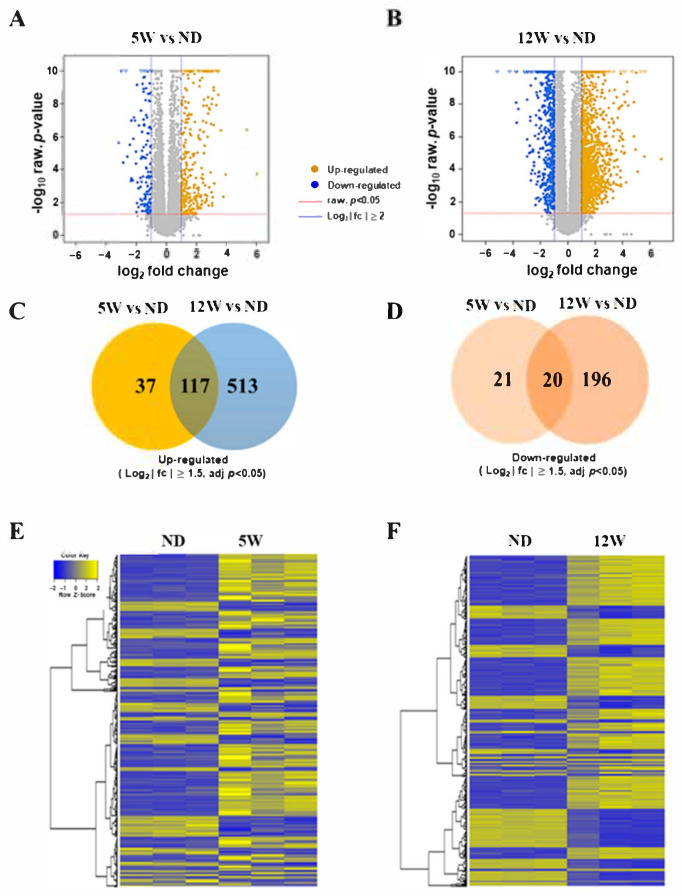
Differentially expressed mRNAs in the renal cortex of the DN rat model. (**A**) Volcano plot for comparison between 5-week-HFD group and normal-diet (ND) group (5 W vs. ND). In the right of the graph, yellow color is indicative of the up-regulated expressed gene (log 2 FC > 2 and FDR < 0.05); in the left of graph, blue color is indicative of the down-regulated gene (log 2 FC > 2 and FDR < 0.05). The gray points indicate mRNAs that were not statistically significant (FDR > 0.05). (**B**) Volcano plot for comparison between 12-week-HFD group and ND group (12 W vs. ND). (**C**) Up-regulated differentially expressed genes (log 2 FC > 1.5, FDR < 0.05) in the comparisons between 12 W vs. ND and 5 W vs. ND. (**D**) Down-regulated differentially expressed genes (log 2 FC > 1.5, FDR < 0.05) in the comparisons between 12 W vs. ND and 5 W vs. ND. (**E**,**F**) Differentially expressed genes (FDR < 0.05) in diabetic rat and normal tissue were analyzed using hierarchical clustering. Each row represents a single mRNA and each column represents one tissue sample. FC, fold change; FDR, false discovery rate; mRNA, messenger RNAs; ND, normal diet; HFD, high-fat diet; 5 W, 5 weeks; 12 W, 12 weeks.

**Figure 5 biomedicines-09-00457-f005:**
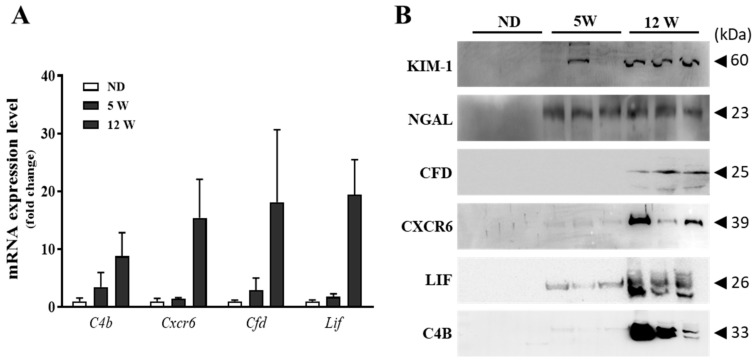
Expression levels of candidate biomarkers in DN rat model. (**A**) mRNA expression levels were determined using qRT-PCR. (**B**) Collected urines were subjected to Western blotting for detecting KIM-1, NGAL, CFD, CXCR6, LIF, and C4B proteins.

**Figure 6 biomedicines-09-00457-f006:**
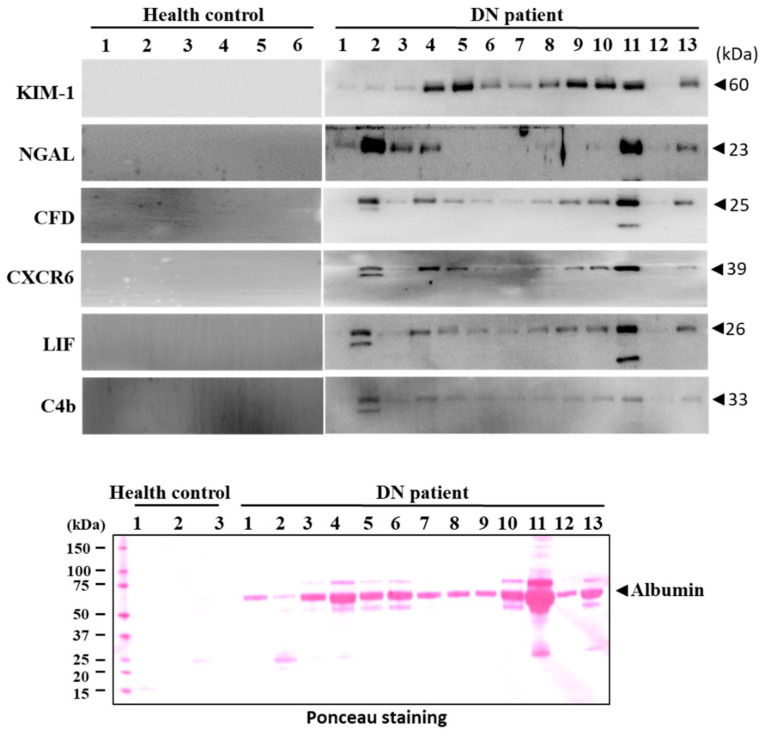
Detection of urinary markers in patients with diabetic nephropathy. Urinary excretion levels of CFD, C4b, LIF, CXCR6, NGAL, TIMP-1, and KIM-1 were measured by Western blot analysis. Membrane was stained with Ponceau O solution to visualize the protein. The major band at about 60 kDa was predicted to be albumin.

## Data Availability

The data that support the findings of this study are available from the corresponding author, upon reasonable request.

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
