# Peer review of "Identification of Novel Biomarker for Early Detection of Diabetic Nephropathy"

_biomedicines, 2021, doi:10.3390/biomedicines9050457_

Round 1
Reviewer 1 Report
In this translational study, KS Kim et al. searched for biomarkers of diabetic kidney disease based on transcriptomic analysis. The research topic is certainly relevant. Validation of biomarkers using proteomic analysis in diabetic rats, as well as in a sample of patients with diabetic nephropathy, is the strength of this study. However, some methodological aspects of the study and its presentation cause some concern. However, some aspects of the study raise questions.
- The sample size is small in both clinical and experimental branches of the study. This needs explanation and should be indicated as a limitation of the study in the Discussion section.
- How was the hypothesis of normal distribution tested? Given the small number of observations in the experimental and clinical groups, the data should be presented as medians, min and max values. Nonparametric parameters would be more reliable in this situation.
- There is no information about participants with diabetic nephropathy and control subjects. The demographic data, albuminuria and renal function parameters should be presented for both cohorts. Besides, type of diabetes and glycemic control parameters are needed for diabetic subjects.
- Searching for biomarkers of diabetic kidney disease is a current topic of investigations. Therefore, it is strange that only few recent articles have been cited in the manuscript.
Author Response
1. The sample size is small in both clinical and experimental branches of the study. This needs explanation and should be indicated as a limitation of the study in the Discussion section.
Answer: Thank you very much for the valuable comment. We agree that the both animal and clinical sample sizes are limited. However, as noted from the previous study, it was clear that KIM-1 and NGAL was considered as markers for diabetic nephropathy in animal study. In Figure 5B, as increase as KIM-1 and NGAL, CFD, CXCR6, LIF and C4B were strongly increased at the 12-week in all urine samples, we believe that suggested molecules can be considered as a candidate. Here, we report this pilot study for the novel finding however, as commented on the sample size, we will increase the clinical sample size for future study.
2. How was the hypothesis of normal distribution tested? Given the small number of observations in the experimental and clinical groups, the data should be presented as medians, min and max values. Nonparametric parameters would be more reliable in this situation.
Answer: As commented, data were presented as medians, min and max values with all points
3. There is no information about participants with diabetic nephropathy and control subjects. The demographic data, albuminuria and renal function parameters should be presented for both cohorts. Besides, type of diabetes and glycemic control parameters are needed for diabetic subjects.
Answer: Based upon diagnostic profile of diabetic nephropathy by clinician’s notes, we used clinical samples. For the clinical urine samples, we basically monitored albuminuria by staining the membrane using ponceau staining method before western blot analysis. Additional figure as added in the Figure 6 (bottom). Health control sample loading is limited due to the lane space.
4. Searching for biomarkers of diabetic kidney disease is a current topic of investigations. Therefore, it is strange that only few recent articles have been cited in the manuscript.
Answer: We added up some recent citation in the Introduction part.
(PMID: 33528734, PMID: 33721945)

Reviewer 2 Report
In this study, the authors evaluated differences in gene expression between non-diabetic rat group and diabetic nephropathy (DN) rat model at various ages (5- and 12-week-high fat induced Zucker diabetic fatty rat) using RNA-Seq. The authors concluded that the transcriptomic data of animal model with progressive diabetes-induced nephrotoxicity indicated that C4b, CXCR6, CFD, and LIF are specific and sensitive biomarker for early detection of diabetic nephropathy.
Comments
The reviewer has some concerns as follows:
- In the methods section, the animal model has some confusing. What is the non-diabetic group (ND)? Is male Zucker diabetic fatty rat fed with normal diets no diabetes induction? Is male Zucker diabetic fatty rat need to be fed with high-fat diet to induce diabetes? This issue should be clarified. The reference(s) can be provided.
- In the methods section, p.110, “with 60% fat” is not correct. “60 kcal% fat” is correct.
- In Figure 2B, the data presentation is confusing. Why blank bars (ND) are also presented in the 5-week- and 12-week-HFD groups?
- In Figure 4, the histological score should be shown.
- In Figure 5, the results are not convincing. In Figure 5A, the statistical analysis is lacking. In Figure 5B, the immunoblots for KIM-1 and LIF in 12-week group should be confirmed and revised. Why there are two bands of immunoblot for LIF existed in 12-week group? In Figure 5C, the IHC images are unclear that need to be revised and quantified.
- The indication for molecular weight should be shown on all immunoblot images.
- The title for supplementary Table 2 (Sequences of primers used in PCR analysis) is not correct.
- The authors concluded that C4b, CXCR6, CFD, and LIF are specific and sensitive biomarker for early detection of diabetic nephropathy. However, the present data cannot support this conclusion. The animal sample size is only 3-4 and not completely convincing. The human samples (n=13) are also not completely convincing.
Author Response
1. In the methods section, the animal model has some confusing. What is the non-diabetic group (ND)? Is male Zucker diabetic fatty rat fed with normal diets no diabetes induction? Is male Zucker diabetic fatty rat need to be fed with high-fat diet to induce diabetes? This issue should be clarified. The reference(s) can be provided.
Answer: Thank you for the valuable comments. Non-diabetic group (ND) was changed to normal diet (ND). Because high fat diet shows diabetic nephropathy in this model shown in the previous report (Amit Kundu, Sachan Richa, Prasanta Dey, Kyeong Seok Kim, Ji Yeon Son, Hae Ri Kim, Seok-Yong Lee, Byung-Hoon Lee, Kwang Youl Lee, Sam Kacew, Byung Mu Lee, Hyung Sik Kim. Protective effect of EX-527 against high-fat diet-induced diabetic nephropathy in Zucker rats. Toxicol Appl Pharmacol. 2020. 390:114899)
2. In the methods section, p.110, “with 60% fat” is not correct. “60 kcal% fat” is correct.
Answer: Corrected.
3. In Figure 2B, the data presentation is confusing. Why blank bars (ND) are also presented in the 5-week- and 12-week-HFD groups?
Answer: Thank you for the comment. We corrected this.
4. In Figure 4, the histological score should be shown.
Answer: we added histological score in the Figure 3B.
5. In Figure 5, the results are not convincing. In Figure 5A, the statistical analysis is lacking. In Figure 5B, the immunoblots for KIM-1 and LIF in 12-week group should be confirmed and revised. Why there are two bands of immunoblot for LIF existed in 12-week group? In Figure 5C, the IHC images are unclear that need to be revised and quantified.
Answer: In Figure 5A, no significant was found because renal injury may not correlate to mRNA level of target gene but leaking the proteins to the urine. Protein can be degraded/aggregated in certain condition. We assumed that LIF protein can be degraded at 12W. Other possible reason is a protein modification. We deleted out the Figure 5C.
6. The indication for molecular weight should be shown on all immunoblot images.
Answer: Corrected. We added molecular weight.
7. The title for supplementary Table 2 (Sequences of primers used in PCR analysis) is not correct.
Answer: Corrected.
8. The authors concluded that C4b, CXCR6, CFD, and LIF are specific and sensitive biomarker for early detection of diabetic nephropathy. However, the present data cannot support this conclusion. The animal sample size is only 3-4 and not completely convincing. The human samples (n=13) are also not completely convincing.
Answer: Thank you very much for the valuable comment. We agree that the both animal and clinical sample sizes are limited. However, as noted from the previous study, it was clear that KIM-1 and NGAL was considered as markers for diabetic nephropathy in animal study. In Figure 5B, as increase as KIM-1 and NGAL, CFD, CXCR6, LIF and C4B were strongly increased at the 12-week in all urine samples, we believe that suggested molecules can be considered as a candidate. Here, we report this pilot study for the novel finding however, as commented on the sample size, we will increase the clinical sample size for future study.
Round 2
Reviewer 1 Report
- A small sample size and the lack of clinical data should be clearly pointed out as the limitations of this study. All results should be considered only as preliminary ones.
- The figures have been changed, but now the statistics in the figures do not correspond to the captions below.
- All statistical symbols in figures must be clearly marked.
Author Response
Reviewer #1
1. A small sample size and the lack of clinical data should be clearly pointed out as the limitations of this study. All results should be considered only as preliminary ones.
Answer: Thank you for the valuable comment. As you pointed out the limitation of sample size, we texted this issue in the discussion part.
2. The figures have been changed, but now the statistics in the figures do not correspond to the captions below.
Answer: Corrected.
3. All statistical symbols in figures must be clearly marked.
Answer: Corrected.

Reviewer 2 Report
The responses for the reviewer's comments can be accepted. No further comments.
Author Response
Thank you very much.